# Pay-as-you-go Population of an Automotive Signal Knowledge Graph

Yulia Svetashova[1], Lars Heling[2], Stefan Schmid[1], and Maribel Acosta[3]

[1] Bosch Corporate Research, Robert Bosch GmbH, Germany
[2] Institute AIFB, Karlsruhe Institute of Technology, Germany
[3] Center of Computer Science, Ruhr University Bochum, Germany

**Abstract.** Nowadays, cars are equipped with hundreds of sensors that support a variety of features ranging from basic functionalities to advanced driver assistance systems. The communication protocol of automotive signals is defined in DBC files, yet, signal descriptions are typically ambiguous and vary across manufacturers. In this work, we address the problem of extracting the semantic data from DBC files, which is then managed in an Automotive Signal Knowledge Graph (ASKG). We developed a semi-automatic tool that automatically extracts signals from DBC files and computes candidate links to the ontology. These candidates can then be revised by experts who can also extend the ontology to accommodate new signal types in a pay-as-you-go manner. The knowledge provided by the experts is stored in the ASKG and exploited by the tool thereafter. We conducted an evaluation of the tool based on a targeted experiment with automotive experts and report on the first lessons learned from the usage of the tool in the context of the Bosch automotive data lake. The results show that our solution can correctly populate the ASKG and that the expert effort is reduced over time.

## 1 Introduction

In automotive engineering, signal processing technologies enable the communication between electronic control units (ECUs) in the car. The ECUs and sensors involved in the communication over the Controller Area Network (CAN) of a car are defined in DBC files[4] A variety of applications rely on logs of the messages transmitted on the CAN bus to analyze and improve the interaction of the ECUs. Therefore, understanding the meaning of the data from DBC files is crucial for subsequent analyses over this data. Moreover, a machine-processable representation of the data semantics facilitates data integration and process automation, which are key requirements for Bosch [14].

Nonetheless, the effective processing of DBC files is not straightforward. The first challenge is associated with the type of identifiers used in DBC files. These identifiers are usually composed of short, abbreviated terms that are difficult to disambiguate. An example of such identifiers is *ACC_Status*, which could refer to the status of the *Adaptive Cruise Control* or the *Automatic Climate Control* component. In turn, the process of extracting the semantics from these identifiers is a challenging task. The second challenge is associated with the heterogeneity of the identifiers. Typically, manufacturers

---

[4] Vector Informatik GmbH. DBC Communication Database for CAN, `https://www.vector.com/`.

use different identifiers to represent the same signals, which makes it difficult to generalize the techniques developed for a specific DBC file.

In this work, we present a novel tool, called CANNOTATOR, which implements a pay-as-you-go approach that combines automatic techniques with human interaction to represent automotive signals defined in DBC files in an Automotive Signal Knowledge Graph (ASKG). At the core of the ASKG is the Vehicle Signal Specification Ontology [8] (VSSo), which is populated and extended with the signals captured in DBC files. CANNOTATOR implements an entity extraction component, that expands the signal identifiers and computes candidate links to the signal classes defined in the ontology. These candidates may be revised by automotive experts before they are added to the ASKG. The goal of ASKG is to establish a Knowledge Base of automotive signals that supports a variety of processes and tools at Bosch in the future. These use cases range from semantic search of datasets in our automotive data lake to entity recognition and linking for NLP in requirements management tools. CANNOTATOR is currently used and evaluated in two pilot projects in different business divisions: Bosch Powertrain and Chassis Systems Control Solutions, prior to a widespread rollout at Bosch.

In this paper, we evaluate CANNOTATOR in the context of the Bosch automotive data lake [14], where we manage in the order of $10^5$ automotive sensor signals from many different projects and test scenarios at Bosch. With CANNOTATOR, experts are able to expand their domain ontology in a pay-as-you-go manner as the need for new concepts arises – without the need of consulting one of the scarce ontology experts. The tool automatically learns from the interactions with the experts and captures their domain knowledge. This continuously improves the tool in being able to process more signals automatically and in assisting experts by providing better recommendations. Our results show that CANNOTATOR effectively assists engineers to expand both the ontology model and the instance data (i.e., signals) in the ASKG.

The remainder of the paper is organized as follows. Section 2 introduces the preliminaries. Our approach is presented in Section 3 and evaluated in Section 4[5]. Section 5 discusses related work and we conclude in Section 6 with an outlook to future work.

## 2   Preliminaries

First, we describe the automotive signal data used as input to construct the ASKG. Then, we present the Vehicle Signal Specification Ontology as the schema of the ASKG.

**Automotive Signal Data.** The Controller Area Network (CAN) is a vehicle bus system used in the majority of today's cars to enable communication between electronic control units (ECUs) that support a variety of features in the car. For the development of complex CAN networks, car manufacturers and their suppliers commonly use the DBC File Format to describe CAN messages. The central object types in a DBC file are *nodes* (i.e., the ECUs), *message*, and *signals*. Listing 1.1 shows an example CAN bus message definition. A message definition starts with the keyword BO_ and is followed by the message identifier (380), message name (POWERTRAIN_DATA), message size (8

---

[5] Developed semantic artifacts, the detailed setup of the evaluation experiment and its results are available at `https://github.com/YuliaS/cannotator`.

Listing 1.1: Example: Message description for steering sensor from a DBC file.

```
1  BO_ 380 POWERTRAIN_DATA: 8 PCM
2   SG_ PEDAL_GAS : 7|8@0+ (1,0) [0|255] "" EON
3   SG_ ENGINE_RPM : 23|16@0+ (1,0) [0|15000] "rpm" EON
4   SG_ GAS_PRESSED : 39|1@0+ (1,0) [0|1] "" EON
5   SG_ ACC_STATUS : 38|1@0+ (1,0) [0|1] "" EON
6   SG_ BRAKE_SWITCH : 32|1@0+ (1,0) [0|1] "" EON
7   SG_ BRAKE_PRESSED : 53|1@0+ (1,0) [0|1] "" EON
8   SG_ CHECKSUM : 59|4@0+ (1,0) [0|15] "" EON
```

bytes) and the node emitting this message (PCM). In the following lines (Line 2 to Line 8 ), the signals that constitute the message are defined. The signal definition starts with the keyword SG_ and is followed by details about the signal. For example, on Line 3 the signal name (ENGINE_RPM) is defined which is followed by additional information, including the starting bit (23) in the message, signal size 16, byte order (0), value type (+), value range ([0|15000]), the unit (rpm) and the intended receiver (EON).

The example reveals some of the syntactic and semantic challenges when trying to automatically match the natural language names to a corresponding semantic model. Engineers use different syntax to delimit words, such as underscores (ENGINE_RPM), hyphens (ENGINE-RPM) or camel-casing (EngineRpm, or EngineRPM). Moreover, they apply different abbreviation methods, such as disemvoweling (PWR = power), acronyms (PCM = powertrain control module) or shortenings (Req = request).

**Vehicle Signal Specification Ontology.** We use the Vehicle Signal Specification Ontology (VSSo) [8] for modeling the ASKG. This ontology is an extension to the Vehicle Signal Specification (VSS),[6] modelled with constructs from the Web Ontology Language (OWL) [11] to express relationships and restrictions. VSSo relies on the SOSA ontology[7] to represent sensors and observations. In this work, we distinguish three main concepts in the VSSo ontology:

*Branch.* In this context, a branch corresponds to a car part. Branches may be structured in hierarchies using rdfs:subClassOf. Examples of top branches in VSSo are *Body*, *ADAS*, and *Cabin*. Branches are associated with signals. These associations are modelled with owl:Restriction on the branch property vsso:hasSignal.

*Signal.* VSSo classifies signals into observable and actuable signals. According to the VSSo specification, the choice of making a signal observable or actuable is based on the existence of the sensor and actuator entries of each VSS signal. In VSSo, signals may be annotated with an abstract definition of the unit using the QUDT[8] ontology, e.g., Length Unit. In addition, VSSo includes human-readable descriptions with the predicates rdfs:label and rdfs:comment. Also, the URIs defined in the VSSo ontology are created with camel case strings for the term at the end.

*Sensor or actuator.* A sensor is a car component that measures a physical variable. An actuator consumes the outcome of sensors to perform actions in the car. In VSSo, the classes sensor and actuator are not necessarily disjoint. Sensors and actuators are associated with signals through restrictions on the signal property sosa:isObservedBy.

---

[6] https://github.com/GENIVI/vehicle_signal_specification

[7] http://www.w3.org/ns/sosa/

[8] https://qudt.org

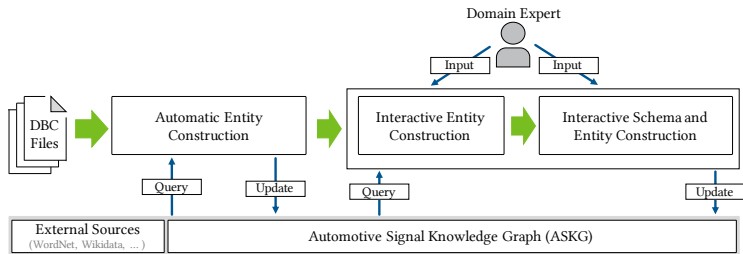

Fig. 1: Overview of the CANNOTATOR architecture

## 3   Automotive Signal Knowledge Graph Population

Given the VSSo as initial schema and a set of DBC files, the problem addressed in this work is to extract the semantics from the signal descriptions provided in the DBC files to populate a knowledge graph (i.e., ASKG) of classes and instances of automotive signals. The ASKG population comprises (i) extending the schema by creating new classes and linking them to existing concepts in the KG, and (ii) creating instances for signals from the DBC file descriptions according to the schema.

We propose a novel semi-automatic tool called CANNOTATOR (cf. Figure 1), which combines automatic techniques with domain expert input to accurately populate an ASKG from the data encoded in DBC files. Our tool processes automotive signal descriptions with the following three main components:

1. **Automatic Entity Construction:** Extracts information from the signal description texts and predicts the signal class of the entity to be added to the ASKG.
2. **Interactive Entity Construction:** Consults experts to create a new signal entity, in the case that a reliable signal class candidate could not be determined automatically.
3. **Interactive Schema Construction:** Extends the ASKG schema with the help of a domain expert, assisted by an adequate Graphical User Interface (GUI).

The central goal of our approach is to minimize the interactions with the domain experts during the ASKG population process. For this, CANNOTATOR operates in a pay-as-you-go fashion such that information provided by the experts is learned by the tool and considered when processing signal descriptions subsequently.

### 3.1   Automatic Entity Construction

This component aims at automatically populating the ASKG by constructing new signal entities from signal description using the existing schema. For a given signal, the meta-data for constructing a new entity is obtained from the descriptions in the DBC File. The component first implements lightweight NLP techniques to the signal name and its unit text to obtain a set of potential expansions, which then map to concepts in the ASKG. Finally, the component uses a complete signal description to identify the corresponding signal class in the KG. During this process, existing information from the KG as well as external data sources are used to aid automatic signal class identification.

*Expanding Signal Names.* The signal names in the DBC files are often composed of several abbreviations or acronyms, which are concatenated using different separators such as camel casing, underscores, or hyphens. Therefore, our tool first splits a given signal name string $s$ into a set of tokens $T_s$. For example, $s$ = "ENGINE_RPM" is split into $T_s = \{$ENGINE, RPM$\}$. For each of the resulting tokens, this component expands the original signal names into full words which can then be matched to identifiers in the ASKG. The expansion relies on knowledge acquired from external sources and that is encoded in the ASKG. Formally, an expanded token is defined as follows.

**Definition 1 (Expanded Token).** *Given a token $t$, an expanded token $e$ for token $t$ is a 3-tuple $e := (t, exp, u)$ with a token string $t$, an expansion of the token string $exp$, and the IRI of the source for the expansion $u$.*

To obtain expanded tokens, CANNOTATOR leverages the KG which contains mappings from abbreviations to expansions that have been created or confirmed by domain experts before. Each mapping in the ASKG is annotated with the IRI of the source. We scraped data on common automotive abbreviations from the Web as the initial abbreviation expansions for the KG. For example, we processed abbreviations provided in Wikipedia[9] and added the expansions to the ASKG. In our example, we obtain the mapping RPM to "revolutions per minute" from Wikipedia and, therefore, we have an expanded token $e_1 = ($RPM, *revolutions per minute*, <https://en.wikipedia.org/>). Since not all tokens are abbreviations but potentially regular words, e.g. ENGINE, we also leverage the WordNet KG[10] to determine whether a token is a word, i.e., $exp = t$. When querying WordNet, we apply lowercasing, lemmatization, and stemming to the token, to increase the chances of finding a correct match.

Since there might be several expansions in the ASKG for an abbreviation, the tool keeps all options for a given token $t$ in a set of expanded tokens $E_t$. For example, a second expansion for RPM might be provided by another source as $e_2 = ($RPM, *rotations per minute*, <http://example.org/car_abbreviations>) and hence, $E_t = \{e_1, e_2\}$. Note that, keeping several possible expansions may increase the chances of finding the corresponding signal class in the KG in the later processing steps. However, if the number of possible expansions is large, it can also become overwhelming and time consuming for domain experts if their input is required. To overcome this, the sources are annotated with trust scores. Formally, we define the trust score as a partial function $\tau : I \mapsto [0, 1]$ that maps an IRI to a trust score value. A higher trust score value indicates higher trustworthiness of the source associated with the IRI and reflects the likelihood of an expansion to be correct. The domain experts can provide feedback on the correctness of these expansions to improve the trustworthiness of the expansions. In this case, the source provenance information is updated to the IRI identifying the domain expert which will have a higher trust score than the Web sources. The experts can also add new expansions to the KG that will be considered when processing future signal names. Lastly, if no expansion could be found, the token itself is considered the expansion and no source is associated with this information,i.e. , $E_t = \{(t, t, \texttt{null})\}$. For example, signal names frequently contain numeric identifiers, which do not need

---

[9] From the article: https://en.wikipedia.org/wiki/Automotive_acronyms_and_abbreviations

[10] http://wordnet-rdf.princeton.edu/

to be expanded. Once this process has been applied to all tokens of a signal name, we construct complete expansions for the entire signal name.

**Definition 2 (Expanded Signal Name).** *Given a signal name s, an expanded signal name $S = \{e \mid e \in E_t, \ \forall t \in T_s\}$ is a set of token expansions for all tokens of the signal name $T_s$. Further, we denote the set of all possible signal name expansions that can be obtained for a signal as $\mathscr{S} = \{S_1, \ldots, S_n\}$.*

In other words, an expanded signal name $S$ is a combination of expansions for each token in the signal name. The set of all possible such combinations for a given signal name is given by $\mathscr{S}$. For our previous example the expansions are[11] $\mathscr{S} = \{$

$S_1 = \{$(ENGINE, engine, `<wn>`), (RPM, revolutions per minute, `<wp>`))$\}$,

$S_2 = \{$(ENGINE, engine, `<wn>`), (RPM, rotations per minute, `<ex>`))$\}\}$

Furthermore, the trust score of an expanded signal name can be defined as the average trust score values of the sources that contributed to the token expansions.

**Definition 3 (Expanded Signal Name Trust Score).** *Given an expanded signal name S, the trust score $\mathrm{T}$ for S is given as $\mathrm{T}(S) := \frac{1}{|S|} \sum_{(t,exp,u) \in S \land u \neq null} \tau(u).$*

*Unit Linking.* After processing the name of the signal, the tool aims at matching the unit of the signal description to the corresponding unit instance in the QUDT ontology. This allows for a better candidate selection later when trying to map the signal to an existing signal class in the KG, as the unit information can help to select the corresponding class. Moreover, in the case that no automatic mapping can be found, this additional information is passed to the experts. Since a variety of unit texts can be encountered and the units in the QUDT ontology typically only have one label (e.g., "Kilometer per Hour" for `qudt:KiloM-PER-HR`), the tool resorts to the Wikidata SPARQL endpoint[12] to retrieve more candidate units. This is possible, since Wikidata links most of the unit instances to QUDT with the "QUDT Unit ID" property (`wd:P2968`) and provides several labels for a single unit (e.g., "km/h","kmh", "kph", "Kilometer per Hour", etc.).

*Signal Class Linking.* Given a set of expanded signal names and links to QUDT unit instances, the approach computes matches for the corresponding signal class in the ASKG schema. This process is detailed in Algorithm 1. The input is a set of signal name expansions $\mathscr{S}$ and a similarity threshold $\theta$. The algorithm iterates over all expanded signal names in $\mathscr{S}$ (Lines 2-15); for each expanded token in a given expanded signal name, a set of candidates from the KG is computed (Line 5). This candidate selection uses a SPARQL query to retrieve the URI ($U$) and the text label ($L$) of all signal classes where the expanded to token $exp$ is either a sub-string of the URI or text label (`rdfs:label` or `rdfs:comment`) from the KG schema. For example, for the expanded signal name $S_2 = \{$(ENGINE, *engine*, `<wn>`), (RPM, *rotations per minute*, `<ex>`)$\}$, we would obtain several candidates for the token "engine" such as `vsso:EngineLoad`, `vsso:EngineOilTemperatur`, `vsso:RotationSpeed`, etc. For each candidate, we then determine a normalized string similarity ($\delta \in [0,1]$) based on the iterative Levenshtein distance between $exp$ and the candidate URI[13] $\delta_U$ (Line 7) and the text label

---

[11] `wn`, `wp`, `ex` stand for the IRIs of WordNet, Wikipedia, and the example source, respectively.

[12] https://query.wikidata.org/

[13] For hash-URIs/slash-URIs we consider the text after the hash/last slash.

---

**Algorithm 1:** Signal Class Linking

---

**Input:** Expanded Signal Names $\mathscr{S}$, Similarity Threshold $\theta$

1. $M = \emptyset$
2. **for** $S \in \mathscr{S}$ **do**
3.     $D$ = empty dictionary
4.     **for** $(t, exp, u) \in S$ **do**
5.        $C = \texttt{candidatesFromKG}(exp)$
6.        **for** $(U, L) \in C$ **do**
7.           $\delta_U = \texttt{similarity}(U, exp)$
8.           $\delta_L = \texttt{similarity}(L, exp)$
9.           $\delta = \max\{\delta_U, \delta_L\}$
10.           **if** $\delta > \theta$ **then**
11.              $D[U] = D[U] \cup \{(exp, \delta)\}$
12.     **for** *(key U, value V)* $\in D$ **do**
13.        **if** $|V| > 1$ **then**
14.           $\bar{\delta} = \frac{1}{|V|} \sum_{(exp, \delta) \in V} \delta$
15.           $M = M \cup \{(S, U, \bar{\delta}, \mathrm{T}(S))\}$
16. **return** $M$

---

$\delta_L$ (Line 8). If the maximum of the similarity values ($\delta$) exceeds the predefined similarity threshold ($\theta$), we consider the candidate an option for the token and add it to a dictionary $D$ that maps the candidate URI to the expanded token and the similarity value (Line 11). After all tokens of an expanded signal name have been processed, we determine whether the same candidate URI has been selected for more than one token (Lines 13-15). The idea is as follows: if several tokens of an expanded signal name map to the same signal class and the computed similarity is high, then it is very likely that this is a correct match. The output of the algorithm is a set $M$ of signal class from the ASKG schema that matches one signal name expansion $S \in \mathscr{S}$. Each match consists of the expanded signal name $S$, the URI of the candidate instance $U$, the confidence of the match $\bar{\delta}$ and the trust score of the signal name expansion $\mathrm{T}(S)$. In our example, both tokens "engine" and "rotations per minute" map to `vsso:RotationSpeed` due to the match with the `rdfs:comment` "Engine speed measured as rotations per minute". Thus, the signal class `vsso:RotationSpeed` is considered a match and is added to $M$.

After all possible classes have been determined for a signal, they are ordered by decreasing similarity $\bar{\delta}$. We use the trust score as a tie-breaker if two matches yield the same similarity. If the similarity exceeds a predefined threshold, we automatically add an instance to the signal class in the KG. Otherwise, the information is passed to the Interactive Entity Construction component. In our example, `ENGINE_RPM` is automatically added to the ASKG using the `vsso:RotationSpeed` class as shown in Listing 1.2.

Listing 1.2: Instance for `vsso:RotationSpeed` and resulting axioms. `vsso-ext` are annotations by the CANNOTATOR to represent the signal in the ASKG

```
1  @prefix askg: <http://www.bosch.com/ns/askg#>.
2  @prefix vsso-ext: <http://www.bosch.com/ns/vsso-ext#> .
3
4  askg:s002  a vsso:RotationSpeed ;
5        rdfs:label "Engine Revolutions Per Minute";
6        qudt:unit unit:RevolutionsPerMinute;
7        vsso-ext:dbcFileName "bmw_i8kk09.dbc";
8        vsso-ext:originalSignalName "ENGINE_RPM";
9        vsso-ext:expandedSignalName "Engine Revolutions Per Minute"@en;
10       vsso-ext:messageName "380Powertrain Data"@en.
```

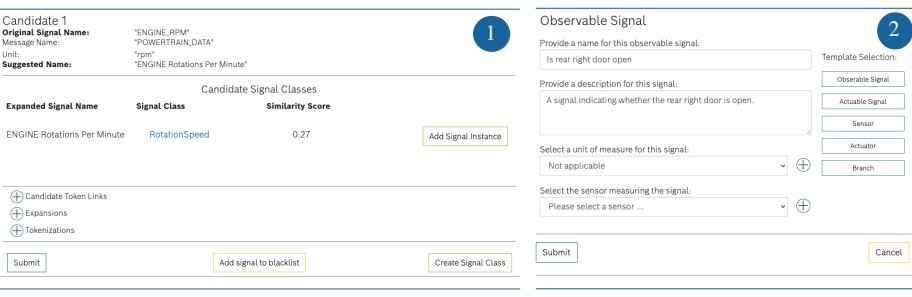

Fig. 2: CANNOTATOR GUIs: 1) Interactive Entity Construction. 2) Interactive Schema Construction generated from Template User Interface (TUI) ontology descriptions

### 3.2 Interactive Entity Construction

The domain experts are only involved to process signals for which no corresponding signal class could be identified automatically. In this case, the experts provide structured input that is added to ASKG through a GUI (cf. Figure 2 (1)). At the top, the original signal name and corresponding message name are displayed as well as the suggested expanded signal name. Below, if there are signal class candidates that match the given expanded signal name (i.e., $|M| > 0$), they are displayed with the corresponding VSSo signal class name and mapping's similarity score. If a correct signal class has not been automatically added due to a similarity score below the threshold $\theta$, the domain expert can directly create the new signal instance by pressing the "Add Signal Instance" button. Further below, there are three expandable sections that can be used by the domain experts to enrich the information about a signal by (i) adding links between tokens of the signal name and other types of classes in the schema (such as branches), (ii) providing expansions for abbreviations, and (iii) providing the correct tokenization of a signal name. With the "Submit" button, the experts can add this additional information to the KG which triggers a re-processing of the current signal description. As a result, new signal class candidates might be generated, so that the signal can be either added automatically to the KG or displayed to the domain expert for further manual processing.

### 3.3 Interactive Schema Construction

If no candidate classes are found in the schema, domain experts can create a new signal class. The Interactive Schema Construction component allows for extending the VSSo through a Graphical User Interface (GUI) with new classes, which are immediately accessible for annotating new instances. The GUIs are generated automatically with an ontology-driven process: Ontology Patterns → Reasonable Ontology Templates (OTTR) → Template User Interface (TUI) ontology → GUI with Input Validation.

*Ontology Patterns.* This component is based on ontology design patterns, defined as "modelling solutions to solve a recurrent ontology design problem" [5]. We use an extended notion of patterns to indicate recurring patterns of axioms in an ontology [9].

Listing 1.3: Pattern *Observable Signal*

```
1  ?classIRI rdf:type owl:Class;
2    rdfs:label ?label;
3    rdfs:comment ?comment;
4    rdfs:subClassOf vsso:ObservableSignal,
5        [rdf:type owl:Restriction;
6        owl:onProperty qudt:unit;
7        owl:allValuesFrom ?unit],
8        [rdf:type owl:Restriction;
9        owl:onProperty sosa:isObservedBy;
10       owl:allValuesFrom ?sensor].
```

Listing 1.4: Class `vsso:RotationSpeed`

```
1  vsso:RotationSpeed rdf:type owl:Class;
2    rdfs:label "Speed"@en;
3    rdfs:comment "Rotations per minute."@en;
4    rdfs:subClassOf vsso:ObservableSignal,
5        [rdf:type owl:Restriction;
6        owl:onProperty qudt:unit;
7        owl:allValuesFrom qudt:AngularVelocityUnit],
8        [rdf:type owl:Restriction;
9        owl:onProperty sosa:isObservedBy;
10       owl:allValuesFrom vsso:RotationalSpeedSensor].
```

Listing 1.5: OTTR Template for *Observable Signal*

```
1  vsso-template:ObservableSignal [owl:Class ?classIRI,
2  rdf:Literal ?label, rdf:Literal ?comment,
3  owl:Class ?unit, owl:Class ?sensor] :: {
4  ottr:Triple(?classIRI, rdfs:label, ?label),
5  ottr:Triple(?classIRI, rdfs:comment, ?comment),
6  o-owl:SubClassOf(?classIRI, vsso:ObservableSignal),
7  o-owl:SubObjectAllValuesFrom(?classIRI,qudt:unit,?unit),
8  o-owl:SubObjectAllValuesFrom(?classIRI,
9          sosa:isObservedBy, ?sensor) }.
```

Listing 1.6: OTTR Instance

```
1  vsso-template:ObservableSignal (
2    vsso:RotationSpeed,
3    "Rotation Speed"@en,
4    "Rotations per minute."@en,
5    qudt:AngularVelocityUnit,
6    vsso:RotationalSpeedSensor
7  ).
8
9
```

The VSSo is both, pattern-inspired and densely interlinked by several recurrent axiom patterns. Firstly, it relies on patterns from the SOSA ontology for modeling sensors, actuators, and observations [8]. Secondly, VSSo was auto-generated from the Vehicle Signal Specification through instantiations of the repetitive axiom structures to define certain types of information (signals, branches, sensors). For example, all signals are defined as subclasses of `vsso:ObservableSignal` and value restrictions on properties `sosa:isObservedBy` and `qudt:unit`. Listing 1.3 shows the pattern for observable signals; leading question marks denote variable elements in the patterns. In our example, `vsso:RotationSpeed` can be instantiated by this pattern as shown in Listing 1.4.

Repetitive structures in VSSo can be captured by seven patterns: *Observable Signal*, *Actuable Signal*, *Observable and Actuable Signal*, *Branch*, *Sensor*, *Actuator*, and *Unit*. We derived these patterns by abstracting from how various classes with the same frequent super-classes are defined in VSSo. To guarantee uniformity and consistency of schema construction, new classes should also become the instantiations of patterns. CANNOTATOR enables that by exposing patterns as templates.

*OTTR Templates.* A template is an abstraction over the underlying pattern. Users of templates create *instances* from templates by providing values to the parameters. These values can be named classes, object and datatype properties, individuals, and plain literals. A template is instantiated by the replacement of its parameters by the provided values. For CANNOTATOR, we used the OTTR [15] OWL vocabulary to record templates and their instances and the instance expansion tool Lutra[15].

Listing 1.5 shows the OTTR template for the *Observable Signal* pattern and the instance of this template for our example class `vsso:RotationSpeed`. Each argument in the instance corresponds to a parameter in a template (e.g., `?classIRI` ←

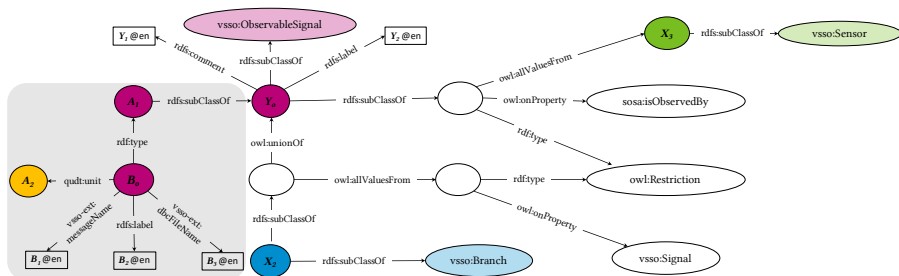

Fig. 3: Template to add new signal classes and instances (indicated in gray) to VSSo

`vsso:RotationSpeed`). The replacement of parameters in a template by instance arguments will generate the class definition given above in the Listing 1.4.

In total, we constructed eight templates: seven – to create classes in the schema according to the mentioned patterns in VSSo and one template to instantiate DBC signal entities (see Fig. 3). For the latter template, the automated entity linking component provides the values for its parameters. For the interactive schema construction, we actively involve the domain experts. Addressing this, we developed a new module of CANNO-TATOR with a graphical user interface that assists the users in completing the templates in a user-friendly and intuitive way based on their domain expertise.

*Template User Interface Ontology and GUI Generation.* Figure 2 (2) shows the graphical user interface of CANNOTATOR with an example of the form for the *Observable Signal* template. Each form element corresponds to a parameter of an OTTR template. The schema-construction process is straightforward: the expert fills in the form, either providing values into text fields or selecting the values from the drop-down menus.

CANNOTATOR implements an ontology-driven generation of this graphical user interface. To this extent, we propose the Template UI (TUI) ontology which maps the elements of the GUI to elements of the OTTR templates. We distinguish three main types of parameters depending on how users interact with them. For `FreeParameters`, users can freely provide values. For `BoundParameters`, a user selects values from a list. `HiddenParameters` are not shown to users and their values are derived from other parameter values. In Figure 3, we denote bound parameters with circles and variables $X_i$, and free parameters with rectangles and variables $Y_j$. Listing 1.7 shows a fragment of the *Observable Signal* UI template: Line 5 represents a hidden parameter for a signal URI, whose value is derived from the class label parameter defined in Line 7. Parameter in Line 12 will be used to select a unit of measurement for this signal from a list.

The parameter descriptions are used to render the GUI form for a template. A field can be either a `ui:TextField` for `Free` parameters or a `ui:Classifier`. For the fields of type `ui:Classifier`, TUI defines the specializations `tui:SingleDropDown` and `tui:MultipleDropDown`, which provide a list of options generated dynamically by evaluating SPARQL queries over the ASKG (e.g., see Line 16 in Listing 1.7).

To minimize the effort of domain experts, CANNOTATOR automatically fills in some of the form fields by exploiting the dependencies between parameters specified by the property `tui:dependsOn` (see, e.g., Line 18 in Listing 1.7). Thus, if a unit of mea-

Listing 1.7: Template UI description for *Observable Signal* template

```
1  @prefix ui: <http://www.w3.org/ns/ui#>.
2  @prefix tui: <http://www.bosch.com/ns/tui#> .
3  @prefix vsso-tui: <http://www.bosch.com/ns/vsso-tui#> .
4
5  vsso-tui:ObservableSignalIRI a tui:HiddenParameter ;
6    tui:processingDirective [vsso-tui:ObservableSignalName, func:mintIRI] .
7  vsso-tui:ObservableSignalName a tui:FreeParameter ;
8    tui:validationFunction func:notEmpty ;
9    tui:parameterGuiFormType ui:TextField ;
10   tui:parameterGuiFormLabel "Provide a name for this observable signal" .
11 vsso-tui:ObservableSignalDescription a tui:FreeParameter .
12 vsso-tui:ObservableSignalUnit a tui:BoundParameter ;
13   tui:validationFunction func:notNull ;
14   tui:parameterGuiFormType tui:SingleDropDown ;
15   tui:parameterGuiFormLabel "Select unit of measurement for this signal" ;
16   tui:parameterFormFillerQuery "SELECT * WHERE {?s rdfs:subClassOf qudt:Unit .}".
17 vsso-tui:ObservableSignalSensor a tui:BoundParameter ;
18   tui:dependsOn vsso-tui:ObservableSignalUnit .
19 vsso-tui:ObservableSignal a tui:Template;
20   tui:parameters ( vsso-tui:ObservableSignalIRI, vsso-tui:ObservableSignalName,
21   vsso-tui:ObservableSignalDescription, vsso-tui:ObservableSignalUnit,
22   vsso-tui:ObservableSignalSensor ) ;
23   tui:ottrTemplate vsso-template:ObservableSignal.
```

surement was selected, the system re-runs a SPARQL query for sensors and pre-selects a corresponding sensor in a drop-down list based on the query evaluation result.

Lastly, CANNOTATOR can change the type of a parameter defined in the TUI descriptions of templates. It happens, for example, if the automated entity generation component obtains a similarity score that exceeds a threshold for the linked sensor or unit entities. Then the parameter's type is set to Hidden and its form field is not rendered.

*Expert Input Validation and Triple Generation.* CANNOTATOR validates the expert input based on the validation functions specified in the TUI templates. These validations can check for empty mandatory fields (e.g., Line 8, Listing 1.7), formatting (date, number, etc.), and potential inconsistencies generated from the data provided by the experts. If the validation fails, CANNOTATOR communicates the reason to the user and asks for correct input. After validation, the tool checks for duplicate concepts by evaluating SPARQL queries against the ASKG. If there exist classes with identical names, labels, or values of restrictions on properties in their class definitions, the user is asked to either (1) confirm that the existing class can fully suit the purpose of signal description (and discard the new class), or (2) change the name or definition of the new class.

For tui:HiddenParameters, CANNOTATOR generates their values by applying processing functions to their source parameters. In our example, it passes the value of the parameter vsso-tui:ObservableSignalName to a function func:mintIRI (see Line 6 in Listing 1.7), which outputs the IRI.

Finally, CANNOTATOR constructs an OTTR instance and runs Lutra to generate triples that are added to the ASKG. In this step, the tool will also extend the ASKG by adding the corresponding signal entity associated with the newly created signal class. The updated ASKG is immediately available to other users or other components.

*Learning from Expert Input.* CANNOTATOR is designed to learn from the input provided by experts with the interactive components. The VSSo extension and the addition

of signal instances to the ASKG, as well as recorded interactions of domain experts with the tool, lead to a continuous improvement of the Automatic Entity Construction component. After each feedback iteration, we say that CANNOTATOR has been *trained*, as more linguistic cues for finding matching signal instances and class candidates become available. In consequence, the manual steps that require a domain expert's attention are naturally reduced over time. In the following section, we will evaluate the usage of CANNOTATOR and focus specifically on the learning aspect.

## 4   Evaluation

We evaluate CANNOTATOR in a controlled setting to empirically study its performance. In a user study, we investigated the usability with 12 experts at Bosch: 5 software engineers with a background in Semantic Technologies and 7 domain experts with a background in automotive engineering. We focused on the following core questions:

**Q1** How well does the template-based schema construction support experts?
**Q2** What percentage of signal names can be handled in a fully automated manner?
**Q3** How does the performance of the system improve over time by learning?

*Input Data.* We randomly selected 200 English signal descriptions out of the 31017 signals from the 82 DBC of different car manufacturers provided by *opendbc*[14]. We excluded object detection data and metadata signals, such as checksums or counters. Two domain experts annotated these 200 signals using our system. Then we selected the 150 signals with the highest inter-rater agreement score as the ground truth. The agreement meant the selection of the same VSSo term for a signal name by both experts or the usage of the same options (superclass, sensor, unit) in the template when the term was missing in VSSo.

*Initial vs. Trained System.* We prepared two CANNOTATOR instances to evaluate the improvement of a *Trained System* over an *Initial System*. The *Initial System*, contained only default knowledge sources in the automated entity construction component and the initial knowledge graph with the VSSo and the QUDT Unit ontology. The *Trained System* contained the input provided by the domain expert from annotating 100 signals of the ground truth dataset. During the annotation process, the KG was extended with the expansions, token relations, alignments, signal instances, and 132 new classes provided by the experts. The KG statistics for both CANNOTATOR systems are shown below.

| System | Triples | Classes | Instances |
|--------|---------|---------|-----------|
| Initial | 20569 | 304 | 0 |
| Trained | 32155 | 436 | 100 |

For the experiments with the users, we created 24 DBC files with 5 signals in each file. The signals for the 12 files to be annotated with the *Initial System* were randomly sampled from the ground truth set of 150 signals. The signals to be annotated with the *Trained System* were sampled at random from the subset of 50 signals not used for training.

---

[14] https://github.com/commaai/opendbc, retrieved on Jul 27, 2020

*Expert Annotations.* Each participant of the experiment was provided with a short introduction to the concept of ontologies, the structure of VSSo, the specifics of the data, and the tool itself. Thereafter, the participants used CANNOTATOR to annotate two DBC files with 5 signal names each. For the first file, they used the *Initial System*, and for the second the *Trained System*. After assessing the candidates provided by the tool, the participants annotate the signals by (1) picking a candidate to *align* the signal, or (2) creating a class to *extend* the ontology using the interactive schema construction component. Note that the input was different for all users because we randomly sampled signals from the subset of the ground truth, which was not used for training the system.

*Evaluation Metrics.* We measured correctness, calculated as the percentage of successfully completed tasks, as the metric for the effectiveness of our approach. Absolute correctness for our tasks is not attainable due to the complexity of the automotive domain and the diversity of CAN-bus data. As the metric for efficiency, we used the time users spent on a task. We report the results for the annotation and the extension separately because the time needed to perform these tasks is different. In summary, we report on the following metrics: (1) **TimeAlign**, mean time a user needed to annotate a signal where the decision was to accept the suggested alignment; (2) **TimeExt**, mean time a user needed to annotate a signal where the decision was to create a new signal class; (3) **TimeAnnot**, mean time a user needed to annotate a signal; (4) **CorrAlign**, percentage of correctly aligned signals (w.r.t. the ground truth); (5) **CorrExt**, percentage of correctly chosen options in the schema construction form; (6) **CorrAnnot**, percentage of correctly annotated signals (aligned or constructed). We compute these metrics separately for the annotations obtained by the *Initial System* and the *Trained System*.

## 4.1   Performance Results

As a result of our evaluation, we obtained 10 annotations from each participant. An annotation could either be an alignment when the user pointed to a VSSo class corresponding to a DBC signal name or a schema extension when the user created a new schema element. The submitted input was compared with the ground truth.

Figure 4 and Table 1 show the performance results for all metrics. As expected, the time for alignment annotations (TimeAlign) is considerably lower than for extensions (TimeExt). For the *Initial System* state, the mean time (in seconds) needed for the alignments was $51s$; extensions took $183.5s$. These averaged into $107s$ per annotation. The mean correctness was $97\%$ for the aligned signals. Schema extensions resulted in $92\%$ correct choices; the correctness of decisions on the level of the signal set was also $92\%$. In the *Trained System*, the majority of the annotations corresponded to alignments; the mean time per alignment was $43s$ (TimeAlign) as well as per annotation in general (TimeAnnot). Only in one case, the user decided to extend the schema, which took $75s$ (left out from Figure 4); this is considerably faster than the average reported for the *Initial System* state. Correctness was $100\%$ for aligned signals in the *Trained System*. Note that we could not compute this metric for the extensions as the ground truth since for the single extensions provided by the domain expert was modeled as an alignment in the ground truth. Yet, this decreased the overall correctness for the annotations to $97\%$.

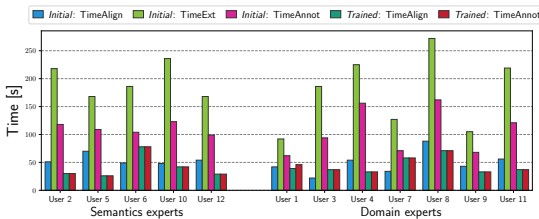

| | System State | |
|---|---|---|
| Metric | Initial | Trained |
| TimeAlign $[s]$ | 50.92 | 42.75 |
| TimeExt $[s]$ | 183.50 | 75.00 |
| TimeAnnot $[s]$ | 107.25 | 43.33 |
| CorrAlign $[\%]$ | 97.25 | 100.00 |
| CorrExt $[\%]$ | 91.67 | NA |
| CorrAnnot $[\%]$ | 91.67 | 97.25 |

Fig. 4: Time spent by each user in the annotations      Table 1: Summary of results

The time spent on schema extensions and their correctness provide insights into **Q1**: in the case that a signal class was not yet present in our ASKG, the experts provided correct template-based schema extensions in a time-efficient manner.

Moreover, our experiment showed that CANNOTATOR effectively learns with expert input: on average, the *Initial System* found alignment candidates for $68\%$ of the signals (for the rest, the users created schema extensions) and the *Trained System* for $100\%$ of the signals. With respect to **Q2**, $28\%$ of the candidates generated by the *Initial System* had similarity scores above $90\%$, which could potentially be handled fully automatically. In comparison, candidates provided by the *Trained System* had higher average similarity scores with $69\%$ above $90\%$ similarity. Yet, human involvement was needed as the tool generated multiple candidates with a high similarity score; in this case, expert input ensures that correct signal instances are added to the ASKG.

Regarding **Q3**, our experiments showed that CANNOTATOR provided more automated alignments in the trained state and, therefore, fewer schema extensions were needed. Overall, the users were two times faster and provided more accurate results using the *Trained System* instead of the *Initial System*.

## 5    Related Work

*Data and Metadata Management Solutions.* Various solutions have been proposed for the large-scale data management of the enterprise data, which offer such functionalities as metadata management and mapping-based data integration (Karma [6], Sansa [10], Ontop [1], or Silk [16]). To the best of our knowledge, none of the existing systems implement schema extension by non-experts combined with the automated mapping candidate generation. These aspects were addressed by several standalone tools and frameworks, which CANNOTATOR builds on.

*Abbreviation Expansion.* The automatic expansion of abbreviations and acronyms has been studied by a variety of works [2,13,19]. Such approaches typically rely on either large corpora to discover abbreviations or leverage the abbreviation's context to determine its expansion. Since the textual data of signal descriptions in DBC files is limited in size and barely provides context, we rely on predefined acronyms and abbreviations which can be extended by the domain experts such that our system improves over time.

*Template-based Ontology Extension Tools and GUIs.* Our system adopts ontology templates to involve domain experts in the schema and entity construction process. Frame-

works and software tools relevant for our approach were developed in the biomedical domain [3,7,12,18]. We could reuse none of the tools directly due to their high domain specificity and limited interaction capabilities. Therefore, we rely on a general template framework called Reasonable Ontology Templates (OTTR) [15] and the tool Lutra[15] for axiom generation. We build template-based GUIs, which is similar to recent works on Web form generation for the interaction with knowledge graphs such as [17] or Shex Form[16]. In contrast to these works that focus on instance data, CANNOTATOR allows for consistently extending the schema of the ASKG.

*Automotive Ontologies.* A variety of ontologies have been developed for the automotive domain. Feld and Müller [4] propose a high-level ontology to describe users, vehicles, and the current driving situation to support Human-Machine Interfaces. Moreover, the W3C Automotive Ontology Community Group[17] proposes vocabularies to improve the interoperability of data in the automotive domain on the Web. However, as these ontologies do not allow for describing ECUs and automotive signals, we use the Vehicle Signal and Attribute Ontology (VSSo) [8] as the schema of the ASKG. The VSSo, which is derived from the Vehicle Signal Specification (VSS)[6], provides a formal model of car signals to improve the interoperability for car development applications.

## 6 Conclusions

We presented a novel tool that assists automotive experts at Bosch in extracting and managing the semantics of CAN signals in an Automotive Signal Knowledge Graph (ASKG). For this, CANNOTATOR implements an entity construction component that automatically extracts signals from DBC files and computes candidate links to the ontology. Experts can then revise these candidates and if necessary extend the schema of the ASKG on-the-fly to accommodate new signal types in a pay-as-you-go manner.

As we have demonstrated through our experiment, CANNOTATOR is capable of learning from the interactions with the domain experts and using this knowledge to improve its assistance capabilities. The results showed that the tool, after some usage by the domain experts, is able to process more signals fully automatically, and also provides higher quality recommendations to the experts. Both lead to a significant reduction in the time that is required by the human experts.

A key lesson learned while designing the automatic entity construction component was the observation that, at first, few restrictions (low threshold) should be applied to the candidate selection. For example, we started by using SPARQL queries with basic string matching filters to obtain candidates from the VSSo, because we found that too restrictive queries (i.e., queries with more triple patterns and constants) would lead to no matches. As the KG grows, the candidates increased, and therefore, the queries can be more restrictive by taking for instance the unit or the branch of a signal into account.

Another lesson learned is that existing template frameworks based on simple tabular interfaces to create template instances [7,15] are not suitable for interactive schema

---

[15] https://gitlab.com/ottr/lutra/lutra

[16] https://github.com/ericprud/shex-form

[17] https://www.w3.org/community/gao/

extension by domain experts. Firstly, they are not capable of providing assistance to the users and involve the experts interactively in the process (e.g. to validate a system recommendation). Secondly, they do not allow to pre-fill the templates with results from the automatic entity construction component or prior input from experts.

Finally, the usage of CANNOTATOR revealed current shortcomings to be addressed in future work: (1) some experts tend to extend the schema instead of spending more time assessing the candidates, and (2) experts sometimes chose inadequate signal classes instead of extending the schema. The key to address these issues is to improve both the ranking of the automatically generated suggestions as well as the usability of the GUIs allowing users to explore the current schema of ASKG during extension.

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
