# OpenReview forum: "Pay-as-you-go Population of an Automotive Signal Knowledge Graph"
_eswc-conferences.org/ESWC/2021/Conference/In-Use_Track — ESWC 2021 In-Use_

### Official Review · AnonReviewer2 · 2021-01-07
**Interesting, but lacking many points from the track call**

**Rating:** 2
**Confidence:** 2

**Review:**

The paper describes a semi-automatic translation of the DBC files used in cars into knowledge graphs. NLP and linking techniques are used where possible, and a GUI over OTTR templates are used for reviewing the automatic suggestions and manual entry of the remaining parts.
The paper is well-written, and easy to follow. The combination of NLP and linking with GUI over OTTR is interesting and I am not aware of anything similar.


**Anonymity:**

No, I would like my review to be deanonymized.

**Strong Points:**

Good explanation of the related data and formats
Detailed explanation of the relation between the original DBC files and the
constructed knowledge graph.
A new GUI for OTTR template instantiation.
An experiment with target end-users.

**Subreviewer:**

I submitted this review.

**Weak Points:**

The paper does not address many of the bullet points in the call of the in-use track:

The actual domain problem solved by this system is not clear.
The problems solved seem to be those introduced by using semantic technology, that is, to populate the knowledge graph and link it to the DBC files

Probably for the same reason, there is also no discussion of existing approaches, no comparison with using
semantic technology, and no description of the impact on the industry.

Although the paper includes an experiment with users in the target group,  there is no description of the current user base, if any.

I see no clear evidence that this application is in use. This criterion is highlighted in the call and must be addressed.

---

> ### Author Rebuttal · Authors · 2021-01-30
>
> Dear reviewer,
>
> Thank you for your feedback on our work.
>
> We start by addressing your comment regarding the usage of our tool. At Bosch, it is common practice to evaluate new tools in pilot projects prior to a wide-spread rollout. In the case of the CANnotator, we have currently 2 pilot projects in different business divisions (Bosch Powertrain and Chassis Systems Control Solutions). As a result, the current user base are automotive domain experts from those divisions that are also expected to use the tool after the pilot phase. With the constructed ASKG, we establish a Knowledge Base of automotive signals that will be used in many processes and tools in the future:
> 1. Semantic search of datasets in our Automotive Data Lake (1. pilot project, see further details here [A])
> 2. In measurement data analytics tools, to allow experts to quickly find and link the correct signals (2. pilot project)
> 3. In requirements management tools, to support NLP/AI algorithms in named entity recognition and linking (under development)
>
> One particular use case is the analysis of logged CAN bus data on a large scale by allowing automotive engineers to query and access this data using the terminology and schema of VSSO and its extension in ASKG. This alleviates the necessity of experts to have information and knowledge about the specific signal names (~500K) in the DBC files. Therefore, our tool provides a semi-automatic approach to construct a schema and instance data that supports ontology-based data access (OBDA) in the data lake.
> We agree that these aspects of the call could be covered in more detail and we will make them more prominent in the camera-ready version.
>
> Regarding the remaining aspects of the call:
> - We provide a description of the problem being addressed, and of its importance in the domain of data integration in enterprise data lakes for automotive data.
> - We describe the components of our tool in detail and indicate how we use Semantic Technologies and how they contribute the tool's functionalities.
> - In the presentation of our tool and the lessons learned, we discuss the challenges associated with the use of Semantic Technologies in the considered scenario. For example, we highlight the challenge of mapping non-semantic data (i.e. the abbreviated and "cryptic" signal names) to the schema of the target ontology. Moreover, we describe the challenge of extracting relevant signal classes from the KG when starting to construct the ASKG (initial system) where few signal classes are available for mapping in the ASKG. Also, we indicate how expert feedback does not only help to populate the KG, but also improves this extraction process by providing relevant metadata to the KG (e.g., abbreviations of domain-specific terms).
> - Finally, we evaluate the performance of our tool according to relevant measures.
>
> [A] "Using Knowledge Graphs to Search an Enterprise Data Lake" (https://link.springer.com/chapter/10.1007/978-3-030-32327-1_46)

---

> > ### Comment · AnonReviewer2 · 2021-01-31
> > **Use base and impact well described**
> >
> > Thank you for your answer, which includes a nice description of use base and more details about impact, which would be sufficient if included in the paper. The call for the In-Use track is and should be narrow, in the sense that there are many questions that are expected to be answered. I realize that your paper does not completely fit the blueprint of the call, as you are addressing issues that come up when applying semantic technology, not a complete solution for some requirement(s). This seems to make the comparison with existing technology not relevant.
> > With the answers, especially about use base, incorporated I suggest accepting this paper.

---

### Official Review · AnonReviewer3 · 2021-01-09
**An annotation tool in the automotive industry, with real-world settings experiments at Bosh**

**Rating:** 1
**Confidence:** 4

**Review:**

This paper describes an annotation tool in the automotive industry, with real-world settings experiments at Bosh. The authors demonstrate throughout the paper how they populate the ASKG graph combining both automatic entity construction and interactive ABox/TBox updates/evolution with experts-in-the-loop. **CANNOTATOR** is evaluated on the learning aspect involving the experts in a controlled setting with three questions involving 7 domain experts.

I found the paper solving a real need, although I missed what types of car applications are currently being developed using DBC and why semantic technologies could replace them with CANNOTATOR. Additionally, it is not clear in the paper which department at Bosch is using/will adopt the tool or if it is supposed to be adopted at the corporate level.

Review details and comments
======================
1. In the introduction, extracting DBC acronyms is challenging. Does that mean there is no existing tool already using DBC files? If yes, how is that tool consumes them to build car applications?
2. Experts in automotive clearly understand the “semantics” of the DBC files. What was difficult to do with an approach consisting of having sessions with them to generate the elements for mappings with VSSo and then use that as an input of the annotator?
3. Page 2. Please define LIDAR. The sentence “The datasets include video, [..] of test vehicles” is difficult to understand. Please, could you rewrite it?
4. I’m curious to understand the use of “pay-as-you-go manner” in their approach. What is the “cost” here?
5. Page 3: To model a car part. Is it possible to use instances of skos:Concept?
6. Figure 1: I suggest to add external sources such as those described in Section 3.1
7. Page 5: In Definition 1, the authors seem to restrict to at most one exp for the expansion u. Would it make sense to allow more than one candidate for exp in u? For trusted sources, since they are fixed sources, I’m curious if at Bosch there is already a policy regarding the use of external sources such as wikipedia, etc.
8. I suggest the use of WordNet 3.1 (wordnet-rdf.princeton.edu) as the reference for WordNet KG.
9.  P.6: I do not understand the use of *iterative levenshtein distance* Does it mean you iterate over the levenshtein distance or it is a new variant of the measure?
10. P.7: What value of the threshold was used in the experiments or set by default per Algorithm 1? Please, mention the value.
11. Add the URI of the namespace ``vsso-ext` in the paper.
12- In Section 3.2 (cf. Figure 2), the experts do not have the *“Semantics”* of RotationSpeed. Do you consider that information relevant to the experts, even if it is written in natural language?
13. In Section 3.3., I’m curious to understand how you manage consensus among experts.
14. More generally, what happens if you enable a reasoner component in the architecture of CONNOTATOR ?
15. P.11: It is not clear to me the difference between TUI templates vs SHACL for validating UI fields. SHACL is a W3C recommendation.
16. P.11: Do you assume that the experts should know/learn OWL in the task of defining a new class? If yes, what is your experience in the learning curve for those experts at Bosch to be able to use the tool?
17. P.11: In ASKG, schema and data are all in the same graph? Would it be easier to explicitly differentiate both concepts by using for example Named Graphs?
18. P.12: Section 4 : What is the impact of the size of the corpus over the results? Let assume you had twice the input data, could that significantly change the current performance results? Please, provide any insight to this respect.
19. P.14: In Figure 4; could it be possible to split the results by emphasizing the results from the domain experts vs software engineers? Maybe that can give more details on how those 2 classes of users actually manage the annotations.
20. Based on the Q3 experiment, could you predict the time when the tool will be fully automatic? How will that affect the human-in-the-loop argument of the tool?


Typos
=====
1. “Our approach is presented in Section 5 “ should be  “Our approach is presented in Section **3**”
2. s/levensthein / **levenshtein**

After rebuttal
===========
Dear authors, Thank you for your detailed comments. I am happy with all the answers to my questions. I understand the process of evaluating tools at Bosch as well as the final users of CANotator. It would have been great to see this work already “in-use” at Bosch (not still in the pilot stage such as (unfortunately) many semantic projects in the industry). I trust that if the paper is finally accepted, the authors will cover in more detail the “in-use aspect” in the camera-ready version so that any reader of the paper will be convinced that Bosch is actually using CANotator.


**Anonymity:**

No, I would like my review to be deanonymized.

**Strong Points:**

* The paper clearly describes the problems when dealing with DBC files in tha automotive domains
* The authors provide a sound evaluation of the tool focusing on the learning aspect from the experts’ input.
* The paper is well organized, easy to read and the architecture of populating ASKG leverages semantic technologies and principles.


**Subreviewer:**

I submitted this review.

**Weak Points:**

* It is not clearly stated in the paper what types of car applications are currently built using DBC files. Thus, what was then the outcomes of designing CANNOTATOR compared to the existing (if any) workflow.
* The paper does not mention which division at Bosch is currently using the tool. Or if it is planned to be deployed at scale in the organization.

---

> ### Author Rebuttal · Authors · 2021-01-30
>
> Dear reviewer,
>
> Thank you for your detailed feedback!
>
> At Bosch, it is common to evaluate new tools in pilot projects prior to a wide-spread rollout. In the case of the CANotator, we have currently 2 pilot projects in different business divisions (Bosch Powertrain and Chassis Systems Control Solutions). The current user base are hence automotive domain experts from those divisions that will also be using the tool after its pilot phase.
> We agree that the in-use aspect of the paper should be covered in more detail and we will make them more prominent in the camera-ready version.
>
> Detailed comments:
> 1. During the development and testing of CAN bus networks in cars, DBC Files are used by engineers to specify messages and signals sent on the CAN bus to enable the communication of ECUs in the car. Therefore, software tools used by engineers during this process make use of DBC Files (e.g. CANoe). However, due to the large variance in ECUs, suppliers, and car models, there exists a plethora of DBC Files with different naming schemas (but potentially describing the same signal). As one of our goals is to use the ASKG to support OBDA for logged data of CAN bus signals from test vehicles, we need to map the terminology (i.e., signal names specified by engineers during the development) to a global schema (VSSo/VSSo-ext).
> 2. The challenge is mapping a large number of (cryptic) signal names to the schema. We estimate the number of signals used in the different variants, product generations, etc. is in the order of 500K. Even experts have difficulties understanding the signals and their relationship in areas besides their field of expertise. Moreover, due to the continuous technology evolution and the increasing number of variants, the task of aligning DBC Files to the ASKG is a never-ending endeavor.
> 3. We will rewrite this sentence and introduce LIDAR.
> 4. In our scenario, the time spent by the experts using the tool can be considered the "cost". As new input data is constantly arriving and due to the semi-automatic nature of the tool, we consider it a "pay-as-you-go" approach.
> 5. In principle, car parts could be modeled with a taxonomy structure using SKOS. However, modelling the particularities of car parts and their relation to the signals on the CANBus (e.g. "Right Door" is related to the signal "IsRightDoorOpen") requires more specific ontological constructs.
> 6. We will add the external sources in the figure.
> 7. At Bosch we leverage external open data sources (if commercial use is allowed) to assist our experts. However, due to the potential quality issues of such data sources, we involve our domain experts in making the final decisions.
> 8. We will update the WordNet reference in the camera-ready version.
> 9. We use an iterative algorithm to compute the levenshtein distance.
> 10. We will add the threshold value of 0.8 to the text.
> 11. The UI of our tool is Web-based and by providing links in the UI, we enable the experts to follow those links and explore the descriptions of the signal name in detail. We will add the VSSo-ext namespace to the paper.
> 12. We assume experts to be specialists. If an expert decides to provide input, we rely on his/her expertise and don't consult other experts for consensus.
> 13. We considered the reasoning capabilities of the ASKG. Concrete signals are added as instances of VSSo signal classes, and several use-case specific features rely on the formal semantics of VSSo.
> 14. Indeed, there are attempts to generate UIs (Schimatos, ShexForm) based on SHACL or Shape expressions, and validate fields. We could reuse none of the tools due to their limited interaction capabilities; CANnotator is highly configurable wrt possible interactions and provides assistance to users. For example, it generates dropdown lists, taking into account the previous choices of a user in the template form, so the queries for options are adapted on the fly. This functionality is not a part of any technologies known to the authors.
> 15. No, we do not assume the experts to have an understanding of OWL and the ontology which is why we have developed the CANnotator (i.e. we assist the experts in expanding the ontology with the need of ontology expertise, following a template-based approach).
> 16. We use different named graphs in ASKG to separate the schema (VSSO, VSSO-ext) and instance data extracted from the DBC Files.
> 17. More input will allow the tool to align more signals automatically since more relevant signal classes will be present in the ASKG and the additional input of the experts will allow for higher trust scores (and the confidence of a correct match) in the automatic component.
> 18.We will highlight the two user groups in the figure.
> 19. We cannot predict the full automation of the tool. The development of new cars will lead to new, potentially unseen input data. Also, as the quality is essential for the ASKG, we don't plan to completely get rid of the human in the loop but reduce their time of interaction.

---

> > ### Comment · AnonReviewer3 · 2021-01-30
> > **Great comments, rating unchanged.**
> >
> > Dear authors,
> > Thank you for your detailed comments. I am happy with all the answers to my questions.  I understand the process of evaluating tools at Bosch as well as the final users of CANotator. It would have been great to see this work already “in-use” at Bosch (not still in the pilot stage such as (unfortunately) many semantic projects in the industry). I trust that if the paper is finally accepted, the authors will cover in more detail the “in-use aspect” in the camera-ready version so that any reader of the paper will be convinced that Bosch is actually using a tool with semantic technology.
> >
> > I maintain my previous rating on the paper and am very happy with the detailed comments from the authors.

---

### Official Review · AnonReviewer5 · 2021-01-14
**Very convincing application and highly potential impact**

**Rating:** 2
**Confidence:** 5

**Review:**

The paper presents a very convincing application  in the automotive industry which is an emerging application domain for Semantic Web. I think it is a timely publication that can inspire several follow-up research papers of Semantic Web publication venues.

The paper is well written with an  insightful understanding of the application requirements and natures of the data. The technical details are not trivial that show authors spending considerable efforts to build the system with several possible technologies/tools. I really like the lessons learnt which can lead to interesting research questions for Semantic Web.

There are few questions/points I think authors could clarify to make the paper more useful to the readers. The first question is what license CANNOTATOR has, I think it'd very great if it has some open source license which will encourage new players in this industry. The second question is about the performance aspect of using RDF data format for CAN data in both annotating and processing/querying phase? Do you have issues with current Semantic technologies/tools?

A minor point, the link to DBC file format should have been put earlier in the paper.

**Anonymity:**

Yes, I would like my review to remain anonymous.

**Strong Points:**

-The paper's application is very convincing

-Paper is well written with insightful technical details

-The paper many potential impacts on Semantic Web research and tools

**Subreviewer:**

I submitted this review.

**Weak Points:**

I see no reason to reject the paper, perhaps, there some technical details can be made clearer as mentioned in the review.

---

> ### Author Rebuttal · Authors · 2021-01-30
>
> Dear reviewer,
>
> We want to thank you for your positive feedback.
> Here is our response to your two questions from the review:
>
> 1. The CANnotator is implemented as a pipeline and is integrated with other systems at Bosch. As such, the code is currently under a closed license. Nevertheless, we have decided to make the generic/reusable components of the tool available (such as the OTTR templates for VSSo, the UI template ontologies, the open CAN files, etc.) as well as those experimental results based on open data in the GitHub repository: https://github.com/YuliaS/cannotator, and will reference it in the paper.
>
> 2. Regarding the performance of the tools we use: So far, we have not experienced limitations with the performance with Semantic technologies by using RDF. We use a commercial-grade KG platform for managing the ASKG (Stardog Graph Database). It should be noted however that we are not transforming the actual CAN bus signal data into RDF. Instead, the ASKG is used to manage and query the metadata extracted from the DBC files to support OBDA.

---

### Official Review · AnonReviewer1 · 2021-01-15
**Interesting and well-developed use-case, but lack of in-use discussion**

**Rating:** 1
**Confidence:** 4

**Review:**

The paper presents a tool, called CANnotator, for translating simple text based technical specifications from the car industry with ambiguous identifiers and unclear semantics into an ontology. The tool uses a variety of tools, techniques and languages to do this: tokenizing, entity extraction, entity linking, ontology patterns and templates, ontology driven GUIs, alignment with existing ontologies, and use of existing data sources such as Wikidata and WordNet. The tool is evaluated by experts as Bosch, where the work is conducted.

I appreciate all the work required to develop the tool and the use-case. The process that the tool sets up, including a human in the loop, seems very sensible and practically useful and should be applicable to many use cases for also other domains.

I have concerns about the novelty of the contribution. The tool seems to me to be an orchestration of many existing and well-known techniques (as alluded to in the end or the first paragraph of the related work section - and hence should rather be called a pipeline or process than a tool?)  The authors claim that the tool is novel, however, it is not clear what the novelty is and what the main contributions of the work are.  Could this be explicitly stated?

The comparison of the selected works in the related work section is quite shallow. The authors makes no real comparison with existing text-to-ontology and ontology learning from text approaches or with other similar use cases.

As for the generality/applicability of the tool/process, it would be interesting to know if and how general the tool is; what would it take to configure the tool to another domain? To what degree is the tool hard-coded for the presented use case?

None of the developed resources (tool executable, ontology, patterns or templates, GUI, evaluation data) from the work seem to be available to the community for reuse or evaluation. Why not?

The paper is predominately a tool description, but given it is an in-use paper the authors could make a better effort at answering the points in the call text https://2021.eswc-conferences.org/call-for-papers-in-use-track/ There is little or no reflection on the quality, use and impact of the *results* of the tool, and the role semantic technologies plays, including for the performance of the tool, e.g.,
 - what is the quality of the resulting ontology?
- did the *semantics* of ontologies used to build the ASKG play   any role?
- "what is the current user base of this system, [...] as well   as plans for deployment/adoption"
- "A discussion on the benefits and challenges associated with   the use of Semantic Technologies ..."

The paper is well written and easy to follow.

Minor things:
 - Undefined abbreviations: DBC, LIDAR
- The description of the VSSo ontology could be shortened
- Figure 3 is difficult to understand.

**Anonymity:**

Yes, I would like my review to remain anonymous.

**Strong Points:**

- interesting use case
- sensible architecture
- towards best practice
- well written

**Subreviewer:**

I submitted this review.

**Weak Points:**

- lack of novelty
- lack of in-use discussion
- tool and resources not available

---

> ### Author Rebuttal · Authors · 2021-01-30
>
> We thank the reviewer for the valuable suggestions and questions.
>
> The novelty of the tool is in the combination of i.) the automated mapping of automotive signal names to known schema elements (with or without human experts in the loop) with ii.) the assisted creation of new entities and schema elements in the automotive domain.
>
> Although several tools exist to support these tasks separately (see, for example, systems for entity linking and knowledge graph population presented in Text Analysis Conference (TAC) [1]; the general template framework OTTR [2]  and tools like TermGenie [3], Populous [4], Webulous [5] – for schema elements creation in the biomedical domain), to the best of our knowledge, none of the known tools provides support in both tasks simultaneously and orchestrates these tools into a coherent workflow (making them mutually beneficial).
>
> Another novel aspect is related to the automotive domain, where both tasks require a high-degree of domain adaptation due to the nature of the automotive signal names used in real systems.
>
> One of the main contributions of this paper is to show how the integrated tool (combining both functionalities) is benefiting domain experts. At Bosch, we evaluate tools in pilot projects prior to a wide-spread rollout of a new tool. In the case of CANnotator, we currently have 2 pilot projects in different business divisions: Bosch Powertrain and Chassis Systems Control Solutions. The exposed experiment results bring insights into the performance of the tool.
>
> While the Call for Paper for the In-Use Track does not require to publish artefacts and source code from our submission, we agree with the comment and decided to make available the generic/reusable components of the tool chain as well as a public version of the ASKG based on experiments with openly available DBC data. (Due to the confidentiality of commercial data, we cannot release our internal ASKG.)
>
> Therefore, we will make the following resources available in the GitHub repository: https://github.com/YuliaS/cannotator, and will reference it in the paper:
> * OTTR templates for VSSo;
> * Template UI ontology for exposing OTTR templates in the UI;
> * The version of the ASKG based on open datasets;
> * DBC files from the snapshot of the project opendbc;
> * Signal list for the experiment with users;
> * Anonymised user performance data.
>
> Due to the limited textual context in the DBC files and the minimalistic descriptions of classes in VSSo, the NLP-based text-to-ontology techniques are hardly applicable to our material. However, we consider it a potentially interesting prospective development for the future use cases, where ASKG is an enabler and serves a reference source for entity linking, for example, for annotating signal names in the texts of requirements.
>
> We will extend the camera-ready paper with the description of use cases, where we use the ASKG as a knowledge base, and highlight the benefits and challenges of using Semantic Technologies in those contexts. In addition, we will add explicit statements answering the following questions:
>
> * “What is the quality of the resulting ontology?”
>     * The templates and the assistance capabilities of the GUI ensure the consistency of the added schema elements.
>     * The human-in-the-loop approach with the interactive GUI for candidate selection contributes to the quality of added signal entities.
>
> * “Did the semantics of ontologies used to build the ASKG play any role?”
>     * Since the ASKG is built by applying ontology templates, which in turn encode patterns extracted from VSSo, the formal semantics of VSSo is considered in the extension of the ASKG;
>     * We use the semantics of VSSo and the QUDT unit ontology to find and rank candidates in the automated signal entity construction.
>
> [1] https://tac.nist.gov/publications/index.html
>
> [2] Skjæveland, Martin G., et al. "Practical ontology pattern instantiation, discovery, and maintenance with reasonable ontology templates." International Semantic Web Conference. Springer, Cham, 2018.
>
> [3] Dietze, Heiko, et al. "TermGenie–a web-application for pattern-based ontology class generation." Journal of biomedical semantics 5.1 (2014): 1-13.
>
> [4] Jupp, Simon, et al. "Populous: a tool for building OWL ontologies from templates." BMC bioinformatics 13.1 (2012): 1-12.
>
> [5] Jupp, Simon, et al. "Webulous and the Webulous Google Add-On-a web service and application for ontology building from templates." Journal of biomedical semantics 7.1 (2016): 1-8.

---

### Official Review · AnonReviewer4 · 2021-01-16
**Interesting usage of VSSo. More resources should be shared. VSS 2.0 has drastically evolved**

**Rating:** 2
**Confidence:** 4

**Review:**

Post-rebuttal comments:

I thank the reviewers for their clarifications and answers. I welcome the new commits https://github.com/YuliaS/cannotator. I wish that the GUI part is also open sourced, is it planned? At the minimum, can screenshots of the tools be inserted in the README with a more complete description? Can you also push the mappings between QUDT and Wikidata (this should be IP free).
I have raised my score to an Accept as I believe this paper presents an interesting use case with a concrete usage of the OTTR template.
-------------

This paper presents an interesting usage of the VSSo ontology. More precisely, it presents an automatic approach supported by appropriate user interfaces for instantiating this model and developing a so-called Automotive Signal Knowledge Graph (ASKG) while extending the schema using data extracted within DBC files and human in the loop validation.

The paper is well written and structured. A number of resources has been developed. However, they are not shared with the community preventing the reproducibility of the results. Hence, is the CANnotator tool publicly available? Can the mappings between QUDT and Wikidata be published (and how accurate are those mappings)? Can vsso-ext be published?

The other major concern I have concerned the evaluation (Section 4) which requires a lot of clarification.
  - The authors indicate that 200 signals have been randomly selected from 82 DBC files. How many signals in total one could find in those 82 DBC files? Since https://github.com/commaai/opendbc is continuously evolving, can the authors share the snapshot of 82 files used, and the exact 200 signals which have been selected?
  - The authors indicate that 150 signals have higher inter-rater agreement ... given that there are 2 annotators, does this mean that the 2 experts fully agree on the VSSo concept to use or the term to add? What about the 50 others?
  - The table shows that the Trained system made used of 132 additional classes: do they correspond to newly created classes?
  - The sentence "The signals to be annotated with the Trained System were sampled at random from the subset of 50 files not used for training" is incorrect. It is not files but signals right?
  - The evaluation puts a lot of emphasis on time to complete the task. The results tend to show that the UI certainly helps but also that simple heuristics enables to match signal mentions with VSSo concepts relatively well.
  - The authors mention an ambiguity problem: given a signal mention in the DBC file, multiple candidates can be retrieved from the ontology. If this is the case, then the problem is actually how to rank the possible candidates so that the best ranked term is the good term. Why not computing a precision @k?

Major comments:
* VSSo (https://github.com/w3c/vsso/) is based on the so-called version 1 of VSS. However, since 1 year, GENIVI is developing VSS 2.0 which has largely been influenced by VSSo. The paper would benefit to comply with the newer version of VSSo in preparation (see e.g. https://github.com/danielwilms/vsso-demo).
* The Automatic Entity Construction is analogous to an entity reconciliation problem., in the sense that the goal is to find from the VSSo ontology which concept should be used for a given signal name / abbreviation. The authors could write this service following the API specification at https://reconciliation-api.github.io/specs/latest/
* In the Algorithm 1, does the "candidatesFromKG" function search in both vsso and vsso-ext?
* It is unclear which tool can process the Template UI presented in the Listing 1.7? Is it lutra?

Minor comments:
* In the introduction, the authors should better explain how the DBC files are generated in the first place and why one can encounter this heterogeneity in how signals are named or abbreviated. Hence, why do DBC files not comply with the VSS standard in the first place?
* In the introduction, the authors claimed that the Bosch data lake contains datasets of 10^5 different automotive sensor signals. Is this due to the many different possible configurations of car models and brands?
* In the introduction: "Our approach is presented in Section 5 ..." -> "Our approach is presented in Section 3 ..."
* The footnote 4 should be moved from the first time DBC is mentioned (line 4 of the introduction)
* The listing 1.1 would benefit to display the actual values (observations of signals at a particular instant). Where are those stored in the DBC files?
* In the Section 3.3, the sentence "In Figure 3, we denote bound ..." would benefit in being in the caption of the Figure 3.

**Anonymity:**

No, I would like my review to be deanonymized.

**Strong Points:**

Interesting usage of the VSSo ontology

Human in the loop approach for enabling both the instantiation of the model in developing the ASKG but also the extension of the ontology.

Make use of ontology design pattern to enforce consistency of the ontology as well as of OTTR to guide the user interface

**Subreviewer:**

I submitted this review.

**Weak Points:**

Many resources have been developed  but they are not shared

The evaluation needs to be clarified (see above)

This work largely relies on VSS 1 while VSS 2 has changed significantly.

---

> ### Author Rebuttal · Authors · 2021-01-30
>
> We would like to thank the reviewer for the insightful comments on the paper, corrections, and the interesting prospective developments towards VSS 2.0.
>
> The CANnotator is implemented as a pipeline and is integrated with other systems at Bosch. We will make some of the developed artefacts and data for the experiment available in the public GitHub repository: : https://github.com/YuliaS/cannotator, and will reference it in the paper.
>
> To address the concerns about the evaluation raised by the Reviewer, we will add the following clarifications to the camera-ready version:
> * The total number of signals in the 82 DBC files, from which we sampled the signals for the experiment, is 31017. Please note that we further excluded non-English signals, object detection data and metadata signals, such as checksums or counters. The snapshot with the set of DBC files along with the list of selected signals is made available in our GitHub repository.
> * For the subset of 200 signals our experts had to either map a signal name to a term in VSSo or create a new signal. Full agreement in the mapping scenario meant the selection of the same VSSo term for a signal name by both experts, in the signal creation case – the usage of the same options (superclass, sensor, unit) in the template to construct a class and the similarity of the textual definition of a signal in the “description” field. The 150 signals with the highest agreement were the signals where experts fully agree on the interpretation of signals. More cryptic or less specific signals (“DTCIUnused1_785”, “prndl”) were those where experts disagreed. We laid these signals aside and did not use signals causing controversy in the experiment with users.
> * The 132 additional classes used by the Trained System were the newly created classes; these classes were not only signals, but also sensors and actuators, used to define newly created signals.
> * We rank candidates by their similarity score and present them in the descending order so that experts can fully assess their relevance by selecting them explicitly in the GUI. However, in the current setting we are not optimising the ranking function, we rather would like to get more input from experts to facilitate the learning capabilities of the system.
>
> Regarding the usage of the newer version of VSSo, please note that this version and the paper “An evolving Ontology for Vehicle Signals” mentioned in the GitHub repository (https://github.com/danielwilms/vsso-demo ), were not available upon the submission of our paper. However, we are thankful for the relevant reference. Our tool can be easily adapted to it. To accommodate the VSSo 2.0, we only need to construct new templates and the corresponding template UI ontology. In contrast to VSSo (1.0), VSSo 2.0 does not use complex class definitions for signal classes. Still, the class naming scheme has not changed much, so we can safely reuse our knowledge bases and algorithms for entity linking and only exchange template sets. We are interested to develop and test them as part of future work.
>
> Addressing the rest of your suggestions and questions will make our paper more readable and we are grateful for your feedback.
> Lastly, we will answer the 4 remaining questions:
> 1. In Algorithm 1, the function "candidatesFromKG" takes into account both VSSo and vsso-ext.
>
> 2. “It is unclear which tool can process the Template UI presented in the Listing 1.7? Is it lutra?”
> The template UI generator and processor is a component of CANnotator. This module dynamically generates form-based GUIs for filling in templates. It provides various assistance functions to users, validates input, and as its output it constructs a valid instance of OTTR template. This OTTR instance is then processed by Lutra. We found out that the dynamic GUI for templates with assistance is crucial for the large scale knowledge graph building in industry context.
>
> 3. “The sentence "The signals to be annotated with the Trained System were sampled at random from the subset of 50 files not used for training" is incorrect. It is not files but signals right?”
> Yes, we will correct it.
>
> 4. “...why one can encounter this heterogeneity in how signals are named or abbreviated. Hence, why do DBC files not comply with the VSS standard in the first place?”
> In the Introduction, we will highlight the use case description and the main challenges related to the signals in the DBC files: car manufacturers (all of them clients of Bosch) use different naming conventions (even different languages) for signals, and there exist thousands of car configurations. The term set of the VSSo alone does not fully cover the domain and all usage scenarios of the DBC data. In the absence of centralized authority for naming and the growing complexity of car applications, VSSo will constantly need extensions to handle the heterogeneity of the DBC data.

---

### Decision · Program_Chairs · 2021-02-23

**Decision:**

Accept

**Comment:**

The reviewers agree that the paper presents a very interesting application of semantic web technologies in a real-world application. The paper is well written and organized, and the approach has been extensively evaluated.
The only remark is about the in-use aspect of the presented approach, i.e., it is not completely clear how the approach is being used, what the target user group is and what is the impact. The authors already resolved some of these remarks in the rebuttal, and it is expected to integrate those changes in the final version of the paper.